# Recursive Inference Scaling: A Winning Path to Scalable Inference in Language and Multimodal Systems

**Ibrahim Alabdulmohsin**
Google Deepmind
Zürich, Switzerland
ibomohsin@google.com

**Xiaohua Zhai**
Google Deepmind
Zürich, Switzerland
xzhai@google.com

## Abstract

Inspired by recent findings on the fractal geometry of language, we introduce *Recursive INference Scaling* (RINS) as a complementary, plug-in recipe for scaling inference time in language and multimodal systems. RINS is a particular form of recursive depth that significantly outperforms +55 other variants, including the recent "repeat-all-over" (RAO) strategy in Mobile LLM (Liu et al., 2024) and latent recurrent thinking (Geiping et al., 2025). Unlike prior works, we carry out our comparisons on a *compute-matched* regime, and demonstrate that for a fixed model size and training compute budget, RINS substantially improves language modeling performance. It also generalizes beyond pure language tasks, delivering gains in multimodal systems, including a +2% improvement in 0-shot ImageNet accuracy for SigLIP-B/16. Additionally, by deriving data scaling laws, we show that RINS improves both the asymptotic performance limits and the scaling exponents. More importantly, with light-weight (linear) adapters (comprising $< 1\%$ of model parameters) and stochastic dropout, RINS offers a *no-regret* strategy, meaning that RINS-enabled pretraining improves performance in language modeling even when recursive depth is not applied at inference time. This corresponds to improving performance on a training compute-, parameter-, and inference-matched regime, suggesting its potential as a viable component of LLM pretraining!

## 1 Introduction

There has been a proliferation of research in recent years pointing to the pivotal role of scale, and how its benefits could be predicted empirically (Hestness et al., 2017; Kaplan et al., 2020; Alabdulmohsin et al., 2022; Bansal et al., 2022; Zhai et al., 2022). Generally, the performance of deep neural networks $f(x)$ (such as its error rate or log-perplexity) often follows a power law $f(x) \sim \beta x^{-c} + \varepsilon$ as one varies a dimension $x$, such as the data size or model parameters. These "scaling laws," as they are known today, have been used, among others, to determine the training data size needed for a specified level of accuracy (Cho et al., 2015; Beleites et al., 2013; Figueroa et al., 2012) and to optimize the model architecture (Kaplan et al., 2020; Hoffmann et al., 2022; Alabdulmohsin et al., 2024b), with some theoretical justification (Bahri et al., 2021; Hutter, 2021; Sharma and Kaplan, 2022).

Besides this conventional approach of scaling training compute, the impact of increased inference compute on model performance has emerged as another key scaling dimension. For example, chain-of-thought (CoT) prompting show that eliciting longer inference paths through additional token generation could improve the reasoning capabilities of LLMs (Wei et al., 2024), similar to the success of critiquing before evaluating (Ankner et al., 2024). Also, AlphaCode (Li et al., 2022) and Codex (Chen et al., 2021) generate multiple samples during inference to enhance code generation. Remarkably, Brown et al. (2024) shows that the benefit of sampling multiple solutions in tasks such

39th Conference on Neural Information Processing Systems (NeurIPS 2025).

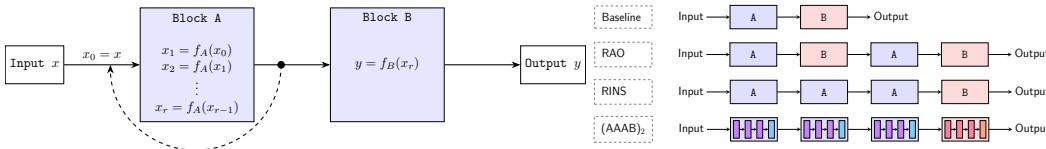

Figure 1: LEFT: In RINS, the model $f : \mathcal{X} \rightarrow \mathcal{Y}$ is split into two parts: the first block $f_A : \mathcal{X} \rightarrow \mathcal{X}$ is applied iteratively to its own output $r$ times before passing the output to the second block. RIGHT: Illustrative examples of models with different signatures and degrees. From top to bottom: (1) *Baseline* (Signature: AB, Degree: 1), a feedforward architecture with no recursion. (2) *repeat-all-over* (RA) (Liu et al., 2024), where the entire model is recursively applied on its output. When recursion is done twice, it has a signature of ABAB. (3) RINS with signature $A^3B$. (4) $(A^3B)_2$ whose degree is 2, in which the same parameter sharing signature is applied on each of the two blocks A and B.

as mathematics and coding—when measured by coverage—holds for up to four orders of magnitude of inference compute. Thus, inference compute follows systematic scaling patterns that can be leveraged to improve models. Refer to the survey by Welleck et al. (2024) for more details.

Recently, it has also been noted that language exhibits a "self-similar" (fractal) nature, meaning that similar patterns repeat across different scales of its representation, from individual words to entire sentences and paragraphs (Alabdulmohsin et al., 2024a). Inspired by this finding, we examine if recursive depth, which can be interpreted as a form of *scale-invariant* decoding, offers a complementary approach for scaling inference time in language models. To this end, we examine an extensive set of parameter-sharing strategies and, indeed, identify the best to be a special form of recursion, which we term *Recursive INference Scaling* (RINS). We show that RINS yields significant performance gains over +55 other methods when controlling for model size and training compute.

RINS builds upon the concept of model recursion but recasts it as a powerful inference-time scaling strategy. It leverages a simple yet profound idea: *use your existing architecture and training compute budget as is, but exploit the self-similar structure of language by recursively applying an early portion of your network to refine its output*. In turn, this simple strategy improves performance significantly.

Recursion has shown promise in language modeling, with recent work by Liu et al. (2024) and Geiping et al. (2025) demonstrating that recursive architectures outperform similarly sized vanilla models trained on a similar number of tokens. However, while such works demonstrate the *sample efficiency* of recursive architectures, in which models are compared when trained on a similar number of tokens, their analysis does not explicitly account for the increased computational cost of recursive operations during training. Hence, it remains unclear whether the performance gains observed in prior work come from the inherent advantages of model recursion or simply from having increased the training compute. Indeed, our findings suggest that for moderately sized models (over 1 billion parameters), the performance gains of "repeat-all-over" (RAO) in MobileLLM (Liu et al., 2024) can be matched by training the baseline model longer to consume an equivalent compute. RINS, by contrast, significantly outperforms all other baselines on a compute- and parameter-matched setup, including when scaling inference by increasing the context length (see Figure 2).

Crucially, a *stochastic* variant of RINS not only can enhance performance further, such as in multi-modal systems, but also provides the flexibility to optionally forgo increased inference computation at test time with minimal performance degradation. We show that combining stochastic RINS with lightweight (linear adapters), comprising $< 1\%$ of parameters, offers a *no-regret* strategy, meaning that RINS-enabled pretraining improves performance in langauge modeling even when recursive depth is not applied at inference time! See Section 4 for details.

We conduct our experiments mostly on compact models, which are typically intended for deployment environments with stringent memory limitations (Liu et al., 2024). Given the direct relation between a model's memory footprint and its parameter count (e.g. a 1 billion parameter model with 16-bit floating-point precision requires 2GB of DRAM), the ability to enhance accuracy while maintaining a fixed parameter count is highly desirable. RINS achieves this by unlocking significant gains without increasing parameter count for the same training compute. In Section 6, we study the effect of sharing the KV cache during recursion to reduce memory footprint even further. While KV cache sharing diminishes some of the gain, RINS with KV cache sharing still enjoys an advantage.

**Statement of Contribution.** In summary, we introduce Recursive INference Scaling (RINS), a complementary plug-in method for scaling inference time. We:

1. propose a taxonomy of parameter-sharing architectures, empirically evaluating their effectiveness. Our comprehensive analysis identifies RINS as a powerful approach, outperforming +55 other methods like RAO used in Mobile LLM (Liu et al., 2024) and latent recurrent thinking (Geiping et al., 2025), and scaling inference by increasing the sequence length.

2. unlike prior works, we control for training compute FLOPs in our comparisons.

3. unlike prior works, we study the effectiveness of recursive depth *beyond language* to multimodal systems that incorporate language in their processing, such as contrastive models. In particular, our SigLIP-RINS-B/16 outperforms the popular SigLIP-B/16 (Zhai et al., 2023) by a wide margin; e.g. improving 0-shot accuracy in ImageNet from 77.3% to 79.6% and CIFAR100 from 70.3% to 80.7%.

4. argue that the performance gain of RINS likely stems from the self-similar (fractal) nature of language, by showing that a similar analysis in vision yields minimal improvements.

5. derive data scaling laws for RINS, revealing improvements in both the asymptotic performance limit and convergence speed (i.e. scaling exponent).

6. show that *stochastic* RINS can enhance performance even further, such as in multimodal systems, while offering the option to revert to non-recursive inference at test time with minimal performance degradation. In particular, with lightweight (linear) adapters, stochastic RINS offer a *no-regret* strategy, suggesting its potential as a viable component of LLM pretraining.

7. analyze the impact of KV cache sharing to reduce memory footprint even further and show that RINS continues to offer an advantage in that setup.

## 2 Recursive Inference Scaling

**Overview.** Before describing how RINS works, we formalize definitions. Let $\mathcal{X}$ be a fixed domain, often the space of sequences of soft tokens of embedding dimension $d$. Let $\mathbb{L} = \{l_1, l_2, ..., l_n\}$ be a fixed set of $n$ unique blocks, where each block $l_k : \mathcal{X} \to \mathcal{X}$ is a function mapping from the input space $\mathcal{X}$ to the same output space $\mathcal{X}$. By "unique" here we simply imply that such blocks (which typically comprise of multiple layers each) are not constrained to share the same parameters. Let $\mathcal{G}(\mathbb{L})$ be the space of all possible computation graphs representing neural network architectures that can be constructed by composing blocks from the set $\mathbb{L}$, while $f \in \mathcal{G}(\mathbb{L})$ be one specific architecture.

Figure 1 (right) illustrates some examples for the case when $|\mathbb{L}| = 2$. For instance, one can repeatedly apply the entire model, as in the "repeat-all-over" (RAO) approach in Mobile LLM (Liu et al., 2024), or recursively apply a strict subset of the architecture, such as a single a block within the model. The choice of arrangement of blocks can significantly impact the model's performance and efficiency.

Formally, let $C(f)$ be the actual computational cost (in FLOPs) of training $f \in \mathcal{G}(\mathbb{L})$ on a dataset sampled i.i.d. from distribution $\mathcal{D}$, considering only the forward pass. Also, $\mathcal{L}(f)$ is a performance metric of interest (e.g., validation loss) for model $f$, with lower values being better.

**Definition 2.1.** For a fixed set of blocks $\mathbb{L}$ and a training compute budget $c$, a recursive architecture $f^\star \in \mathcal{G}(\mathbb{L})$ is called "better" than another $f \in \mathcal{G}(\mathbb{L})$ if $C(f^\star) \leq C(f) \leq c$ and $\mathbb{E}[\mathcal{L}(f^\star)] \leq \mathbb{E}[\mathcal{L}(f)]$.

In other words, we search for the architecture $f^\star$, constructed only from the set of blocks $\mathbb{L}$, that minimizes the loss under the constraint of a bounded training compute $c$.

Model recursion offers a simple, plug-in approach for scaling inference time. By applying some layers iteratively to their own output, we effectively increase the computational path length during inference without altering the underlying model architecture. This allows us to exploit the benefits of increased inference compute. Importantly, it is *complementary* to other techniques like chain-of-thought (CoT) prompting (Wei et al., 2024) and repeated sampling (Li et al., 2022; Chen et al., 2021).

**Taxonomy.** As discussed in Section 1, language has been shown to exhibit self-similarity, meaning that similar patterns repeat across multiple levels of granularity, from the structure of individual sentences to the composition of entire texts (Alabdulmohsin et al., 2024a). This observation suggests that recursive (scale-invariant) decoding could be particularly beneficial for language processing.

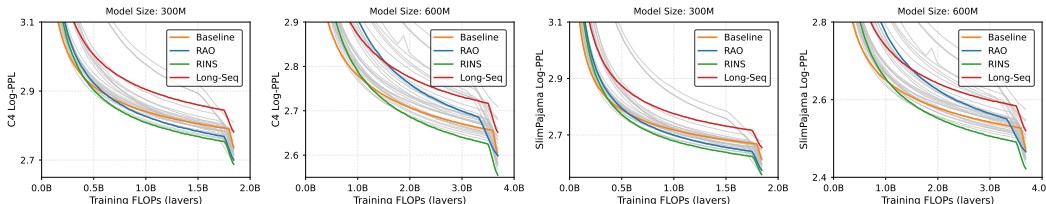

Figure 2: Language models are trained on 200B tokens. The $x$-axis is the training cost in units of layer $\times$ step. Notably, the performance advantage of RINS increases with longer training. The long-sequence baseline, using a context length of 1,536 tokens, exhibits lower performance due to processing fewer examples to maintain the same FLOPs count. See Figure 3a for longer training durations and Figure 4 for larger (1B parameter) models, further demonstrating the value of RINS. Sharp drops in perplexity near the end of training are due to learning rate cooldown.

| Model Size (# params) | Recursive Signature | # examples / core / sec ($\uparrow$) | Peak Memory Usage (GiB) ($\downarrow$) |
|---|---|---|---|
| 300M | Baseline (`AB`) | 22.7 | 2.60 |
| | RINS (`A`$^2$`B`) | 18.1 | 3.31 |
| | RINS (`A`$^3$`B`) | 13.9 | 3.56 |
| | RAO (`ABAB`) | 12.6 | 4.35 |
| 600M | Baseline (`AB`) | 12.0 | 3.92 |
| | RINS (`A`$^2$`B`) | 8.5 | 5.70 |
| | RINS (`A`$^3$`B`) | 6.4 | 6.31 |
| | RAO (`ABAB`) | 6.5 | 7.48 |

Table 1: A summary of latency and memory footprint of recursive architectures. We observe that RINS with `A`$^2$`B`signature has a significantly better latency and memory footprint than RAO and still outperforms it significantly in terms of quality.

To determine the extent to which this is true, we systematically explore a vast space of parameter sharing strategies. We use the following taxonomy based on two key concepts: (1) *signature* and (2) *degree*. The "signature" of an architecture describes how parameter sharing is applied across different blocks of the model. For example, an architecture with the signature `AB`$^2$`C` indicates that the model is partitioned into three unique blocks (`A`, `B`, and `C`) of *equal* size. The processing flow would be: apply block `A` on the input, apply `B` to `A`'s output, apply `B` again to its *own* output (hence the exponent), and apply `C` to the output of `B`. Note that while the signature `AB`$^2$`C` has the same parameter count as `ABC`, it incurs $\approx 33\%$ more compute due to the repeated use of block `B`. This highlights the trade-off between computational cost and performance using recursion that we will carefully study in this work.

On the other hand, "degree" specifies if recursive depth is nested, as illustrated in Figure 1 (bottom). A degree of 2, for example, means that each of the blocks `A`, `B`, and `C` in our example have themselves the same recursive pattern used in the overall model. This adds another dimension to our taxonomy, allowing for a finer-grained classification of recursive architectures. Degree makes notation concise, but is not necessary; e.g. (`ABB`)$_2$ is equivalent to `ABB CDD CDD`. By systematically varying both signature and degree, we can comprehensively explore the space of recursive models and identify optimal configurations. Appendix A provides a detailed pseudocode. With this framework, we now state our main question: *Which family of architectures (i.e. signatures and degrees) lead to better performance according to Definition 2.1 under fixed compute budget?*

To reiterate, this is a non-trivial question because it is possible for a non-recursive model to outperform all others given that it sees more training examples, since we always match from FLOPs. For instance, increasing the context length can be inferior to longer training within a compute constraint, as shown in Section 3. Nevertheless, our analysis reveals that *Recursive INference Scaling* (RINS) emerges as a clear winner. Its consistent superiority suggests that it captures a fundamental principle for efficiently processing language. We hypothesize that this is due to the self-similar geometry of language.

**Definition 2.2.** RINS corresponds to architectures with degree 1 and signature `A`$^r$`B`, for some $r > 1$.

In other words, RINS partitions a model depthwise into two equally-sized blocks `A` and `B`. The first block `A` is recursively applied on its own output $r$ times before applying `B`. See Figure 1 for illustration and Appendix A for a detailed pseudocode. In Section 6, we study the optimal value of $r$.

| | BL | RAO | RINS | | | BL | RAO | RINS |
|---|---|---|---|---|---|---|---|---|
| OpenBookQA | $37.1 \pm 0.8$ | $\mathbf{40.0 \pm 0.6}$ | $39.0 \pm 0.3$ | BoolQ | | $53.1 \pm 4.7$ | $58.3 \pm 0.6$ | $\mathbf{59.5 \pm 0.8}$ |
| PIQA | $65.8 \pm 3.4$ | $68.8 \pm 0.5$ | $\mathbf{69.8 \pm 0.3}$ | SIQA | | $40.2 \pm 0.5$ | $40.0 \pm 0.5$ | $\mathbf{40.9 \pm 0.4}$ |
| HellaSwag | $46.0 \pm 1.0$ | $49.6 \pm 0.3$ | $\mathbf{50.2 \pm 0.5}$ | CommonSenseQA | | $29.0 \pm 1.6$ | $31.3 \pm 1.7$ | $\mathbf{32.4 \pm 1.0}$ |

Table 2: 0-shot evaluation in downstream common sense reasoning tasks. All models are 600M parameters, pretrained on the compute-equivalent of 500B tokens in the baseline (BL). The best signatures for RAO and RINS in Figure 3a are used in this evaluation. See Appendix C for details.

**Main Claim.** Our claim can be summarized as follows. Once a language model is chosen and the training compute budget is planned, one should enable RINS during training, which does not change the model size, and train the new RINS-enabled model for the *same* amount of compute. Our empirical results demonstrate that RINS-enabled models will consistently outperform baseline models. In addition, stochastic RINS, particularly with linear adapters (see Section 4), offers the option of forgoing scaling inference at test time with little performance degradation.

## 3 Experimental Results

In this section, we study the impact of various parameter-sharing strategies in language modeling, following the taxonomy introduced in Section 2. We show how RINS emerges as a clear winner. All experiments are carried out using the Big Vision codebase (Beyer et al., 2022).

**Setup.** First, we train a decoder-only transformer language model (Vaswani et al., 2017) with relative position embeddings and sequence packing. We use C4/English tokenizer with a vocabulary size of 32K. The model is trained on a mixture of C4 (Raffel et al., 2020) and SlimPajama (Soboleva et al., 2023) with equal weight using a batch size 1,024 and context length 1,024. The non-recursive baseline is trained for 200K steps, which amounts to about 200B training tokens. Other recursive models are trained on *fewer* tokens in order to *match* the same total compute FLOPs.

The optimizer is Adam (Kingma and Ba, 2014) with learning rate $5 \times 10^{-4}$ and weight decay $5 \times 10^{-5}$, using an inverse square root decay schedule with 5K warm-up and 5K cool-down steps. This learning rate was chosen by sweeping across the values $\text{lr} \in \{10^{-3}, 5 \times 10^{-4}, 10^{-4}, 5 \times 10^{-5}\}$ with $\text{wd} = \text{lr}/10$. The reported value yielded the best performance for the non-recursive baseline. The full training configuration along with a list of all the architectures we sweep across (i.e. signature and degree) can be found in Appendix E. Each model is trained on $16 \times 16$ TPUv5 chips for approximately 2.2K core-hours. Models that failed OOM were excluded. Overall, we train 59 models of two sizes: 300M and 600M parameters, which include RAO (Liu et al., 2024) and latent recurrent thinking (Geiping et al., 2025) whose signature is $\text{AB}^r\text{C}$. In all models, the embedding dimension is 1,536 and the MLP dimension is 6,144. In Figure 3a, we present results using models with 1 billion parameters.

**Results.** Figure 2 shows that RINS outperforms all other approaches. Importantly, the optimal number of recursion rounds in RINS (e.g. whether to use $\text{A}^2\text{B}$ or $\text{A}^3\text{B}$ or more recursions) depends on the allocated training compute budget. This is seen in Figure 2 in the fact that the non-recursive baseline $\text{AB}$ (a special case of RINS with $r = 1$) initially outperforms all other models, before its performance saturates and recursive models begin to outperform it for the same training compute FLOPs and parameter count. We study the relation between the optimal number of recursion rounds and compute later in Section 6. In addition, RINS outperforms scaling inference time by only increasing the sequence (context) length (green vs. red curves in Figure 2). A summary of latency and memory footprint is provided in Table 1.

**Longer Training Duration.** One observation in Figure 2 is that the performance gap seems to widen in favor of RINS as more training compute is allocated. To verify this, we resume training the 600M-parameter models on 500B training tokens. Here, we only compare RINS with the non-recursive baseline and RAO, since RAO was identified in prior works to outperform other parameter-sharing strategies (Liu et al., 2024). Figure 3a shows that the performance gap continues to increase in favor of RINS. This improvement also persists *downstream*, as shown in Table 2.

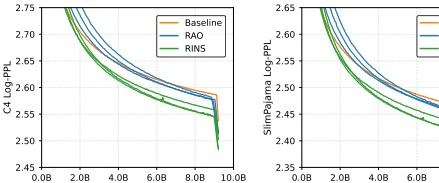 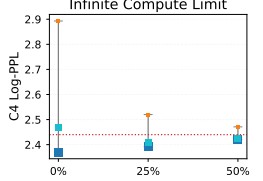 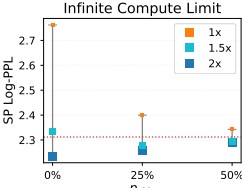

(a) Performance of RINS ($A^2B$, $A^3B$, $A^4B$) vs. RAO ($A^2$, $A^3$, $A^4$) and baseline plotted against increasing compute budget on 600M-parameter models. The performance advantage of RINS grows with the computational budget. The long-sequence baseline is not shown since it underperforms other methods in Figure 2.

(b) Asymptotic performance of 1B-parameter LMs, evaluated in C4 (left) and SlimPajama (right) using *stochastic* RINS. Dotted line is for baseline. We observe a tradeoff between inference flexibility (gap between 1x & 2x) and potential gain of inference scaling (2x result). We resolve this tradeoff in Section 6.

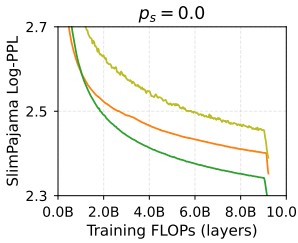 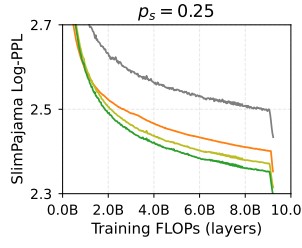 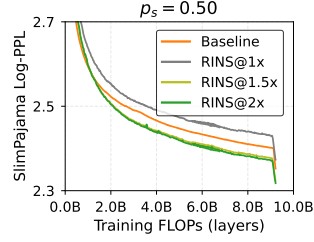

Figure 4: Performance of stochastic RINS ($A^3B$) with varying inference costs for 1B parameter LMs. The $x$-axis represents the training compute cost. The legend indicates the inference cost of each stochastic RINS configuration relative to the baseline; e.g. $1.5x$ denotes 50% increase in inference cost. For $p_s = 0$, RINS@1x is significantly worse, with perplexity scores $> 3$. As expected, RINS converges in performance to the baseline as $p_s \to 1$. Similar results using C4 are in Appendix F.

## 4 Stochastic Recursive Inference Scaling

Next, we investigate the effect of stochastically *dropping* blocks during training, inspired by the regularization technique of stochastic depth (Huang et al., 2016). Our primary goal is to determine whether this approach can further enhance the performance of RINS while simultaneously offering the flexibility of *reverting* to non-recursive inference without significant degradation in model quality.

To recall, RINS has the signature $A^rB$ for some $r > 1$. To implement stochastic RINS, we introduce a skip probability $p_s \in [0, 1)$ and sample during training the number of recursion rounds in each step to be $1 + \eta$, where $\eta$ is a binomial random variable with probability of success $1 - p_s$ and number of trials $r - 1$. Thus, block $A$ is always executed at least once. During inference, we are free to choose how to scale compute by setting the value of $r \geq 1$. See the detailed pseudocode in Appendix A. For this, we train bigger models with 1 billion parameters. We use an embedding dimension 2,048 and MLP dimension 8,192. All models have 18 decoder blocks. We train for 500K steps and compare RINS with signature $A^3B$ against the non-recursive baseline.

Figure 4 summarizes the advantages of stochastic RINS. Notably, we observe that as $p_s > 0$ increases, stochastic RINS mitigates the performance degradation incurred when scaling is not applied at inference time, while still offering big gains when inference time is scaled. Not surprisingly, though, scaling inference time is less effective when $p_s$ increases, suggesting a tradeoff between flexibility at inference time and the magnitude of potential gains from scaling. As shown in Figure 3b, similar conclusions hold in the asymptotic (infinite-compute) regime, assuming the loss follows a power law relation (Kaplan et al., 2020). We resolve this apparent tradeoff in Section 6 using linear adapters.

## 5 Multimodal Systems

**Setup.** Besides language, we study the impact of RINS in vision-language pretraining, motivated by the fact that such models also process natural language. We pretrain SigLIP-B/16 models, which are contrastive models trained using the sigmoid loss on English-language image–text pairs (Zhai et al.,

|  | Dataset | AB | Long-Seq | ABAB | AAB |
|---|---|---|---|---|---|
| *0-shot classification* | ImageNet | 73.4 | 73.0 | 72.7 | **74.1** |
| | CIFAR100 | 68.9 | 63.6 | 65.3 | **72.2** |
| | Pet | 90.4 | 90.0 | **90.7** | 90.0 |
| *Retrieval* | COCO img2txt@1 | **62.7** | 62.0 | 61.1 | 62.3 |
| | COCO img2txt@5 | **84.8** | 84.1 | 82.9 | 84.1 |
| | COCO img2txt@10 | **90.7** | 90.4 | 89.5 | 90.2 |
| | COCO txt2img@1 | 44.6 | 44.2 | 43.2 | **45.1** |
| | COCO txt2img@5 | 69.6 | 69.4 | 68.4 | **70.0** |
| | COCO txt2img@10 | 78.8 | 78.7 | 77.5 | **79.0** |
| | Flickr img2txt@1 | **89.6** | 87.2 | 87.9 | 88.9 |
| | Flickr img2txt@5 | 98.0 | 97.8 | 97.5 | **98.5** |
| | Flickr img2txt@10 | 99.1 | 98.9 | 98.9 | **99.3** |
| | Flickr txt2img@1 | **75.1** | 73.7 | 73.2 | 74.3 |
| | Flickr txt2img@5 | 92.3 | 92.1 | 91.5 | **92.4** |
| | Flickr txt2img@10 | 95.6 | 95.6 | 95.6 | **95.8** |

Table 3: Performance of SigLIP-B/16 on ImageNet (Deng et al., 2009), CIFAR100 (Krizhevsky, 2009), Pet (Parkhi et al., 2012), COCO (Chen et al., 2015), and Flickr (Young et al., 2014). All models are identical in size to SigLIP-B/16 and have the same training compute FLOPs. Long-Seq is SigLIP-B/16 trained on higher resolution of 280 and text length 80 (25% increase in sequence length $\rightarrow$ 50% increase in inference cost, similar to AAB). Wilcoxon signed rank test (Wilcoxon, 1992), gives $p = 0.003$ so the evidence in favor of RINS is statistically significant at the 99% confidence level.

2023). We follow Zhai et al. (2023) in most aspects. Images are resized to $256 \times 256$ with $16 \times 16$ patch sizes. Texts, on the other hand, are tokenized using C4 tokenizer (Raffel et al., 2020) with a vocabulary size of 32K, and we keep a maximum of 64 tokens. The optimizer is Adam with learning rate $10^{-3}$ and weight decay $10^{-4}$, using an inverse square root decay schedule with 5K warm-up and cool-down steps. For the baseline, we use SigLIP-B/16 pretrained on 10B training examples. Again, recursive models are trained on fewer examples to match the total training compute cost. Due to the amount of compute involved in these experiments, we only compare the non-recursive baseline (with signature AB) against RAO (with signature ABAB) and RINS (signature $A^2B$) with degree 1.

**Results.** Table 3 shows that RINS (with signature $A^2B$) outperforms the non-recursive baseline, including with long sequence length, and RAO in zero-shot and retrieval evaluations. Of importance is the impact in ImageNet 0-shot classification, where we see an improvement of about $+0.7\%$.

**Overtraining.** Next, we demonstrate that RINS can substantially advance state-of-the-art results in multimodal systems for a given model size in the overtraining regime. We train a recursive variant of SigLIP-B/16, denoted SigLIP-RINS-B/16, using the signature $A^3B$. In light of the findings presented later in Section 6, we increase the number of recursions here given the increase in compute.

We adhere to the training protocol outlined above, with the exception that SigLIP-RINS-B/16 is now trained on 40B examples, matching the training data scale of the widely-used, publicly available SigLIP-B/16 checkpoint. Note that both models have an identical size. Moreover, following Pouget et al. (2024), we utilize a training mixture comprising both English and multilingual data to enhance cultural diversity, and report the cultural metrics recommended in Pouget et al. (2024) as well as multilinguality evaluations using XM3600 dataset (Thapliyal et al., 2022). So, to ensure an appropriate comparison of results, we re-train SigLIP-B/16 on 40B examples from the same data mixture. The primary datasets for cultural diversity evaluation are Dollar Street (Rojas et al., 2022), GeoDE (Ramaswamy et al., 2024), and Google Landmarks Dataset v2 (GLDv2) (Weyand et al., 2020). We use the Gemma tokenizer (Mesnard et al., 2024).

As shown in Table 6, SigLIP-RINS-B/16 significantly outperforms the standard SigLIP-B/16 across all benchmarks. In fact, *stochastic* RINS with skip probability $p_s = \frac{1}{4}$ improves results even further. Crucially, these results demonstrate that RINS offers a fundamental advantage in multimodal learning that are not replicated by simply overtraining a non-recursive counterpart.

# 6 Further Analysis

**Vision.** As previously discussed, the performance gains in RINS are consistent with the self-similar nature of language. By performing a recursive, scale-invariant decoding, RINS introduces an inductive

| | Dataset | SigLIP-B/16 | SigLIP-RINS-B/16 | | |
|---|---|---|---|---|---|
| | | $p_s = 0$ | $\frac{1}{4}$ | $\frac{1}{2}$ | |
| 0-shot classification | ImageNet | 77.3 | 79.0 | **79.6** | 79.2 |
| | CIFAR100 | 70.3 | 78.5 | **80.7** | 81.7 |
| | Pet | 92.6 | **94.8** | 94.4 | 92.7 |
| Cultural diversity | GeoLoc:Dollar Street | 17.6 | **19.7** | 19.2 | 19.3 |
| | GeoLoc:GeoDE-Country | 22.5 | 23.3 | **24.7** | 23.9 |
| | GeoLoc:GeoDE-Region | 36.1 | 37.7 | **39.7** | 38.7 |
| | Dollar Street | 51.5 | 52.9 | **53.1** | **53.1** |
| | GeoDE | 93.1 | 93.7 | **94.3** | 94.2 |
| | GLDv2 | 51.6 | **53.7** | 52.4 | 52.5 |
| Multilinguality | XM3600 img2txt@1 | 48.4 | **53.5** | 52.6 | 51.8 |
| | XM3600 txt2img@1 | 39.5 | 43.0 | **43.1** | 41.9 |
| Retrieval | COCO img2txt@1 | 67.6 | 69.4 | **70.0** | 69.5 |
| | COCO txt2img@1 | 50.3 | 51.5 | **52.4** | 52.0 |
| | Flickr img2txt@1 | 91.9 | 92.9 | **93.5** | 92.4 |
| | Flickr txt2img@1 | 80.1 | 80.7 | **81.4** | 80.5 |

Table 4: Performance of multilingual SigLIP models on various datasets under an overtraining regime. All models are identical in size to SigLIP-B/16. As shown in the rightmost columns, stochastic RINS ($p_s > 0$) outperforms the other models. Full retrieval & multilinguality results are in Appendix B.

bias that encourages the model to recognize and exploit recurring patterns at different scales (see Appendix D for further discussion). To test if this is likely the source of its advantage, we conduct a similar empirical evaluation in vision, a domain lacking self-similarity. Appendix G provides the full evaluation results using encoder-only vision transformers (ViT) (Dosovitskiy et al., 2021) on ImageNet-ILSRCV2012 (Deng et al., 2009). We observe that recursive architectures, including RINS, do not confer any advantage in the supervised image classification domain, in agreement with our hypothesis that relates the success of RINS to fractal structure of language.

**Data Scaling Laws.** Next, we investigate the influence of Recursive INference Scaling (RINS) on the data scaling behavior of language models. Specifically, we fit a power law of the form $\varepsilon(x) = \beta x^{-c} + \varepsilon_\infty$, to the log-perplexity loss $\varepsilon(x)$ as a function of the training FLOPs $x$. This allows us to analyze the impact of RINS on both the scaling exponent $c$ and the asymptotic performance limit $\varepsilon_\infty$, revealing whether the performance gains in RINS stem from an improved scaling exponent, a lower asymptotic limit, or a combination of both. We use the 600M-parameter language models in Figure 3a whose signature is $\mathtt{A}^r\mathtt{B}$, for $r \in \{1, 2, 3, 4\}$, where $r = 1$ corresponds to the non-recursive baseline. As shown in Figure 5a, RINS improves *both* the scaling exponent and the asymptotic limit. The improvement in the asymptotic limit provides further evidence that the performance gap in favor of RINS is not closed by overtraining non-recursive models.

**Optimal Number of Recursion Rounds.** We observe from the scaling parameters in the previous section that if the performance of RINS is modeled as a power law $f_r(x) = \beta_r x^{-c_r} + \varepsilon_r$, then $c_r$ increases with $r$ while $\varepsilon_r$ decreases. Furthermore, the coefficient $\beta_r$ also increases with $r$. This implies that while scaling inference by increasing $r$ might initially exhibit a higher loss due to the larger $\beta_r$, its faster convergence ($c_r$) and lower asymptotic limit ($\varepsilon_r$) will eventually lead to superior performance. In other words, using the higher recursion level is advantageous eventually, which is consistent with the experimental results. To quantify this more explicitly, we train language models with four signatures $\mathtt{A}^r\mathtt{B}$: $r \in \{1, 2, 3, 4\}$. Then, we plot the optimal value of $r$ against training compute. As shown in Figure 5b, the optimal value of $r$ monotonically increases with training compute, in agreement with earlier results. Also, *smaller* models benefit *more* from RINS.

**Adding Linear Adapters.** Earlier in Section 4, we showed that enabling stochastic RINS during training exhibits a tradeoff between worst-case and best-case performance, depending on whether or not RINS is applied at inference time. Next, we introduce a additional improvement: when a maximum of $r$ recursion rounds are used in stochastic RINS, we add $r$ lightweight, linear adapters (i.e. linear projection layers) to the output before the projection head. The choice of which adapter to apply is a function of how many recursion rounds are used. Specifically, if signature $\mathtt{A}^r\mathtt{B}$ is used, the $r$-th adapter is applied. Empirically, this introduces $< 1\%$ more parameters and has a negligible impact on FLOPs. Yet, it resolves the tradeoff encountered earlier as shown in Figure 6 (left). With

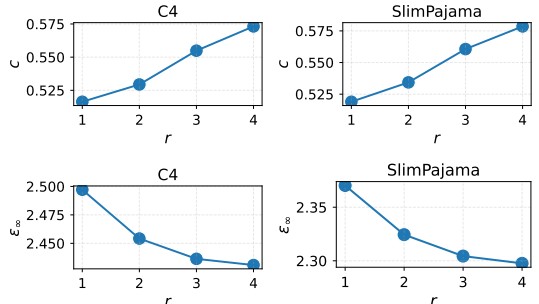
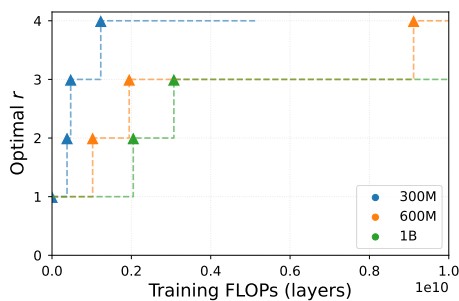

(a) TOP: Scaling exponent $c$ in models with signature $\mathtt{A}^r\mathtt{B}$, where $r = 1$ is the baseline. RINS ($r > 1$) improves scaling exponents. BOTTOM: Asymptotic log-perplexity $\varepsilon_\infty$ is plotted against $r$. RINS improves $\varepsilon_\infty$, so overtraining the baseline cannot match its performance gain.

(b) Optimal number of recursions $r$ in RINS is plotted vs. training compute for 300M-, 600M-, and 1B-parameter LMs. Compact, overtrained models benefit more from RINS.

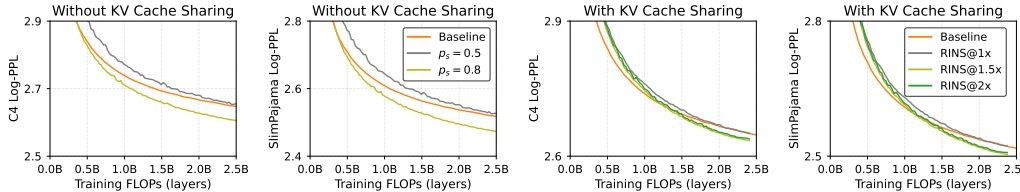

Figure 6: LEFT 2 PLOTS: $y$-axis corresponds to performance when RINS is enabled during training but *disabled* at inference time in 600M-parameter LMs. $p_s = \frac{1}{2}$ in stochastic RINS with linear adapters matches the baseline at $1\times$ the inference cost while $p_s = 0.8$ results in a *better* language model, even though all models have the same training compute, parameter count, and inference cost. We speculate this is because RINS provides a better inductive bias. RIGHT 2 PLOTS: RINS with $p_s = 0.5$ improves performance with KV cache sharing, although the improvement diminishes.

linear adapters, stochastic RINS can provide a *no-regret* strategy, where performance improves in RINS-enabled pretraining even when RINS is not applied at test time.

**Adding KV Cache Sharing.**    Finally, RINS by itself reduces the memory footprint by matching performance using small models, which is important for compact models as discussed earlier. We now push this memory reduction further by analyzing the effect of KV cache sharing, where queries during recursion attend to the tokens and keys of the first block in RINS. Specifically, for models with signature $\mathtt{A}^r\mathtt{B}$, in every subsequence use of block $\mathtt{A}$, queries attend to the cached keys and values of the first call of $\mathtt{A}$. This ensures that memory footprint does not grow with recursive depth. Figure 6 shows that while KV cache sharing diminishes the gain, RINS with KV cache sharing still offers an advantage compared to the baseline.

## 7   Discussion and Related Works

The premise that extending inference time enhances the quality of language model outputs finds support in cognitive science. Human deliberation, often associated with System 2 thinking, is linked to improved decision-making capabilities (Lawson et al., 2020). Mirroring this, numerous computational strategies have been developed to scale inference within Large Language Models (LLMs). Prompt-based methods like Chain-of-Thought (CoT) (Wei et al., 2024) and critique-then-generate approaches (Ankner et al., 2024) are prominent examples. Other iterative refinement methods like ReAct (Yao et al., 2023), Self-Refine (Madaan et al., 2024), and Reflexion, which incorporate feedback and reflection, also improve inference quality. Even simpler strategies, capitalizing on the stochastic nature of LLM decoding that generate multiple responses and select among them, such as self-consistency (Wang et al., 2023), have shown efficacy that scales well across multiple orders of magnitude of inference compute (Brown et al., 2024).

However, the assumption of monotonic improvement with increased inference calls is not guaranteed to hold. As argued by Chen et al. (Chen et al., 2024), repeated sampling may lead to convergence towards the most probable, but not necessarily the optimal, solution. This is particularly pertinent for challenging instances where the probability of a correct answer is below chance (i.e., $< 0.5$).

In this work, we introduce a complementary approach, called Recursive INference Scaling (RINS), which can be integrated with other techniques. RINS identifies a specific form of model recursion as an effective inference scaling strategy, demonstrating significant performance gains over various other recursive architectures, including latent recurrent thinking (Geiping et al., 2025) and the Repeat All Over (RAO) strategy identified by Liu et al. (2024) as state-of-the-art for mobile LLMs.

Recursion is a form of parameter sharing, a technique that has been explored for a while. Illustrative examples include Universal Transformers (Dehghani et al., 2019) and ALBERT (Lan et al., 2020). Despite their innovative design, their scaling exponents were smaller than in the vanilla transformer (Tay et al., 2022a) so they failed to subsume it (Tay et al., 2022b). RINS, by contrast, improves both the scaling exponents and asymptotic performance limits.

## Acknowledgement

The authors would like to thank Alexander Kolesnikov, André Susano Pinto, Michael Tschannen, Jeremiah Harmsen, Olivier Bachem, Jamie Lohoff, Jean-bastien Grill, Mathieu Blondel, Michaël Sander, Tianlin Liu, and Vincent Roulet, for the insightful discussions. We also thank the Google DeepMind unit at large for providing a supportive research environment.

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

# A Recursive Inference Scaling (RINS) Pseudocode

```python
class RecursiveBlock():
  config: Dict  # single block config
  signature: str = "a"
  degree: int = 1
  p_skip: Tuple[float, ...]]  # skip prob for blocks

  def __call__(self, x):
    """Call model on input x."""
    if degree == 1:
      blocks = {c: SingleBlock(config=self.config
      ) for c in set(self.signature)}
    else:
      blocks = {c: RecursiveBlock(
        config=self.config, signature=self.signature,
        degree=self.degree - 1, p_skip=self.p_skip
      ) for c in set(self.signature)}
    inputs = x
    # forward pass
    for i in range(len(self.signature)):
      c = self.signature[i]
      choice = random.uniform()
      if self.degree == 1:  # stochastic RINS
        if choice > self.p_skip[i]:
            x = blocks[c](x)  # else skip
      else:
        x = blocks[c](x)
    return x
```

Figure 7: Numpy-like syntax for models with a fixed signature and degree. When no stochastic depth is applied, we have $p_s = 0$. In RINS, we expand $p_s$ into a tuple of the form $(0, p_s, p_s, \ldots, p_s, 0)$, where the first and last entries are zero to guarantee they are executed, which is equivalent to sampling the number of recursion rounds from a binomial distribution as described in Section 4.

# B  Retrieval and Multilinguality Detailed Results

The following tables provide the full retrieval and multilinguality results for SigLIP-RINS-B/16.

| Language | Image-to-Text Retrieval @ 1 | | | | Text-to-Image Retrieval @ 1 | | | |
|---|---|---|---|---|---|---|---|---|
| | SigLIP-B/16 | SigLIP-RINS-B/16 | | | SigLIP-B/16 | SigLIP-RINS-B/16 | | |
| | | $p_s = 0$ | $p_s = \frac{1}{4}$ | $p_s = \frac{1}{2}$ | | $p_s = 0$ | $p_s = \frac{1}{4}$ | $p_s = \frac{1}{2}$ |
| ar | 54.4 | 58.8 | 59.0 | 58.2 | 42.2 | 45.8 | 46.7 | 45.3 |
| bn | 7.0 | 11.2 | 10.3 | 8.7 | 4.8 | 7.2 | 6.8 | 5.1 |
| cs | 51.1 | 55.8 | 55.8 | 54.5 | 40.5 | 45.0 | 44.6 | 43.0 |
| da | 61.3 | 68.8 | 68.8 | 65.2 | 45.8 | 51.6 | 52.6 | 49.8 |
| de | 81.5 | 84.1 | 83.3 | 83.3 | 69.3 | 72.2 | 72.4 | 71.5 |
| el | 38.8 | 45.8 | 44.9 | 42.9 | 27.4 | 32.1 | 32.2 | 31.5 |
| en | 55.6 | 56.7 | 56.0 | 56.3 | 53.2 | 53.4 | 53.4 | 53.2 |
| es | 67.9 | 71.3 | 71.7 | 70.4 | 62.7 | 63.7 | 63.5 | 63.9 |
| fa | 54.4 | 61.3 | 58.6 | 59.6 | 48.1 | 51.6 | 51.6 | 51.9 |
| fi | 34.6 | 43.2 | 43.0 | 39.8 | 21.9 | 28.4 | 28.8 | 25.9 |
| fil | 19.0 | 21.8 | 20.5 | 19.4 | 10.4 | 12.5 | 12.3 | 11.3 |
| fr | 74.8 | 77.6 | 76.2 | 76.9 | 67.0 | 69.5 | 69.9 | 68.0 |
| hi | 20.6 | 26.5 | 25.9 | 23.5 | 10.5 | 14.6 | 14.4 | 12.3 |
| hr | 46.9 | 57.4 | 56.2 | 53.1 | 33.6 | 41.2 | 40.4 | 37.9 |
| hu | 47.6 | 55.2 | 53.9 | 53.9 | 37.4 | 42.7 | 43.1 | 41.2 |
| id | 74.1 | 77.8 | 77.7 | 78.4 | 65.3 | 68.0 | 68.2 | 67.0 |
| it | 73.9 | 79.1 | 77.5 | 77.3 | 67.3 | 70.4 | 70.2 | 69.9 |
| iw | 48.2 | 56.8 | 54.5 | 52.3 | 37.5 | 43.9 | 43.1 | 40.8 |
| ja | 48.8 | 57.8 | 57.9 | 53.9 | 38.1 | 43.2 | 42.0 | 39.6 |
| ko | 58.6 | 65.6 | 63.3 | 63.3 | 50.0 | 53.4 | 52.5 | 51.7 |
| mi | 0.7 | 0.7 | 0.8 | 0.6 | 0.2 | 0.4 | 0.3 | 0.3 |
| nl | 61.8 | 67.8 | 65.0 | 66.2 | 55.1 | 58.8 | 59.0 | 57.7 |
| no | 61.0 | 68.9 | 67.1 | 66.8 | 45.5 | 52.5 | 52.0 | 50.9 |
| pl | 62.3 | 68.9 | 67.4 | 67.1 | 54.0 | 57.6 | 59.0 | 57.1 |
| pt | 69.6 | 71.1 | 70.1 | 71.4 | 61.5 | 62.6 | 63.3 | 63.2 |
| quz | 7.2 | 7.7 | 7.9 | 7.7 | 3.1 | 3.0 | 3.2 | 2.9 |
| ro | 52.7 | 61.4 | 60.5 | 60.0 | 39.8 | 48.5 | 48.3 | 46.0 |
| ru | 66.7 | 70.8 | 69.5 | 70.3 | 61.3 | 64.2 | 64.3 | 64.2 |
| sv | 66.8 | 71.9 | 71.9 | 70.1 | 51.3 | 55.3 | 56.6 | 54.4 |
| sw | 9.2 | 10.2 | 10.0 | 9.4 | 4.6 | 5.2 | 5.4 | 4.4 |
| te | 1.1 | 1.4 | 1.6 | 1.0 | 0.5 | 0.5 | 0.7 | 0.5 |
| th | 29.5 | 38.2 | 35.1 | 34.5 | 21.5 | 24.7 | 23.1 | 24.0 |
| tr | 52.9 | 58.0 | 57.9 | 57.6 | 44.3 | 47.3 | 46.9 | 46.6 |
| uk | 52.2 | 58.1 | 56.1 | 56.6 | 38.7 | 44.3 | 44.9 | 42.8 |
| vi | 73.8 | 79.3 | 78.4 | 77.8 | 61.8 | 65.7 | 66.4 | 65.5 |
| zh | 55.3 | 60.2 | 60.5 | 58.7 | 44.9 | 48.3 | 49.8 | 47.1 |

Table 5: Per-language retrieval evaluations using the Crossmodal-3600 dataset Thapliyal et al. (2022).

| Metric | SigLIP-B/16 (370M params) | SigLIP-RINS-B/16 (370M params) | | |
|---|---|---|---|---|
| | | $p_s = 0$ | $\frac{1}{4}$ | $\frac{1}{2}$ |
| *Multilinguality* | | | | |
| XM3600 img2txt@1 | 48.4 | **53.5** | 52.6 | 51.8 |
| XM3600 img2txt@5 | 68.4 | **72.6** | 72.0 | 70.9 |
| XM3600 img2txt@10 | 74.4 | **78.1** | 77.5 | 76.5 |
| XM3600 txt2img@1 | 39.5 | 43.0 | **43.1** | 41.9 |
| XM3600 txt2img@5 | 59.5 | 63.0 | **63.1** | 61.6 |
| XM3600 txt2img@10 | 66.1 | **69.3** | **69.3** | 68.0 |
| *Retrieval* | | | | |
| COCO img2txt@1 | 67.6 | 69.4 | **70.0** | 69.5 |
| COCO img2txt@5 | 87.2 | 88.6 | **88.9** | 88.3 |
| COCO img2txt@10 | 92.6 | 93.2 | **93.5** | 93.2 |
| COCO txt2img@1 | 50.3 | 51.5 | **52.4** | 52.0 |
| COCO txt2img@5 | 74.7 | 75.2 | **76.0** | 75.7 |
| COCO txt2img@10 | 82.6 | 83.0 | **83.5** | 83.2 |
| Flickr img2txt@1 | 91.9 | 92.9 | **93.5** | 92.4 |
| Flickr img2txt@5 | 99.3 | 98.7 | **99.5** | 98.7 |
| Flickr img2txt@10 | 99.6 | 99.5 | **99.7** | 99.3 |
| Flickr txt2img@1 | 80.1 | 80.7 | **81.4** | 80.5 |
| Flickr txt2img@5 | 94.6 | 94.3 | **95.4** | 95.0 |
| Flickr txt2img@10 | 97.1 | 97.0 | **97.4** | 97.0 |

Table 6: Performance of multilingual SigLIP models on various datasets under an overtraining regime. All models are identical in size to SigLIP-B/16. As shown in the rightmost columns, stochastic RINS ($p_s > 0$) outperforms the other models. Full retrieval & multilinguality results are in Appendix B.

## C   Zero-shot Evaluation on Common Sense Reasoning Tasks

This section describes the evaluation setup for assessing the zero-shot common sense reasoning capabilities of the language model (Table 2).

**Datasets.**   Each model is evaluated on six benchmarks:

- **OpenBookQA**: A multiple-choice question answering dataset requiring knowledge from an open book of elementary level science facts (Mihaylov et al., 2018).
- **BoolQ**: A yes/no question answering dataset requiring the model to determine the truthfulness of a given statement based on a provided passage (Clark et al., 2019).
- **PIQA**: A multiple-choice benchmark focusing on physics-related reasoning, such as how we interact with objects in daily life (Bisk et al., 2020).
- **SIQA**: A multiple-choice dataset focusing on social common sense reasoning (Sap et al., 2019).
- **HellaSwag**: A multiple-choice dataset focusing on sentence completion (Zellers et al., 2019).
- **CommonSenseQA**: A multiple-choice question answering dataset with distractors (Talmor et al., 2019).

**Prompting.**   Each model is evaluated in a zero-shot setting, meaning it does not receive any training examples specific to the task. Instead, since all tasks above have multiple-choice answers, we evaluate the log-perplexity of each option, applying a causal mask, after concatenating the contextual information (if any), followed by the prefix and the answer. Then, we select the choice that has the lowest per-token log-perplexity score as the model's answer.

We do not apply any prompting, except in BoolQ and PIQA. In BoolQ, we use the prompt template:

```
<data['passage']> Based on this, the answer to the question:
<data['question']>, is: ...,
```

which we have found to improve performance significantly.

In PIQA, we formulate each sentence in the form:

```
The goal is: {goal} The solution is: {sol}.
```

We do this in PIQA because, otherwise, the sentences can be difficult to understand, even for humans. In one example, for instance, the goal is: "Deep clean coffee grinder." and the two possible solutions are: "Scrape with rice." and "Scrape with flour." Concatenating directly would result in sentences like "Deep clean coffee grinder. Scrape with rice." whose are unclear.

**Evaluation Metric.** The evaluation metric is accuracy. For each example, the model's predictions are compared to the ground truth labels, and the accuracy is calculated as the percentage of correct predictions. The model is evaluated in a deterministic mode, meaning no randomness is involved in the inference process.

# D   Recursion and Self-Similarity

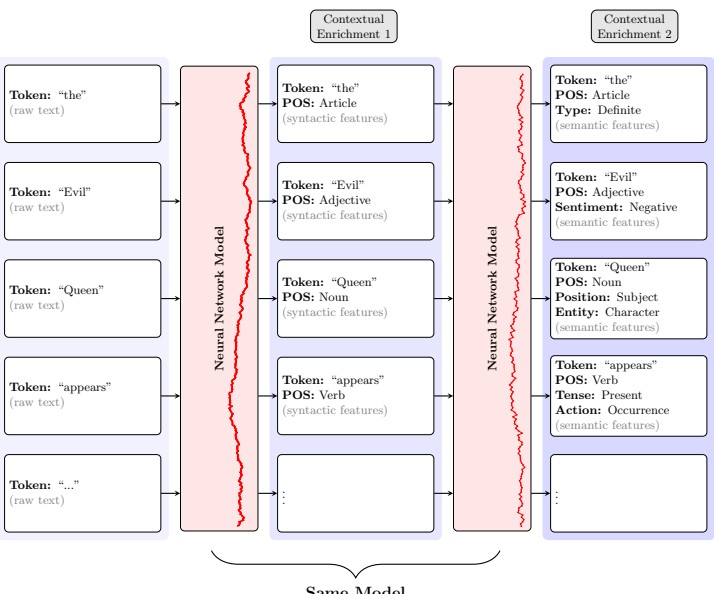

Figure 8: Caption

How does depth-wise model recursion relate to self-similarity of language across the temporal dimension? We provide an illustration of this in Figure 8.

Informally, one might view the early layers in a neural network to be transforming the raw linguistic input into a *richer language* that is appended with useful, high-level details. For instance, in the sentence "the evil queen appears ...", early layers potentially through mechanisms like attention, might modify the token "queen" to incorporate features such as part-of-speech (POS) tagging, sentiment, and grammatical information. Subsequent layers can then iteratively incorporate progressively higher-level contextual information.

The concept of self-similarity in temporal sequences, including language, hints that higher-order linguistic features exhibit patterns analogous to those present in the raw input. If this property holds for language, it implies that the *same* neural network block can be recursively applied to generate increasingly abstract features.

However, while this interpretation offers a conceptual framework, it is essential to acknowledge that it remains a hypothesis. Empirical evidence has demonstrated the self-similarity of log-perplexity scores in language (i.e. surprise of information-theoretic complexity), indicating that patterns observed at the word level are mirrored at sentence and potentially higher levels of linguistic structure. The efficacy of Recursive INference Scaling (RINS) is another evidence in favor of the informal picture above.

# E  Configuration

## E.1  Architecture Sweep

We use the following pseudocode when sweeping across recursive architectures. The goal is to ensure that we only compare architectures that have a similar size.

```python
# baseline
add(signature='A', degree=1)

# RAO
add(signature='AA', degree1)
add(signature='AAA', degree1)
add(signature='AAAA', degree1)

# other architectures
for signature in ['ABB', 'ABA', 'AAB', 'ABBC', 'AABC', 'ABCC', 'ABBB', 'AAAB', 'AABB']:
    for degree in [1, 2, 3]:
        num_unique_blocks =  len(set(signature) ** degree unique_layers
        num_layers_per_block = num_layer_in_baseline_model // num_unique_blocks
        if num_layers_per_block > 0:
            add(signature=signature, degree=degree)
```

## E.2 Training Configuration

### E.2.1 Language Modeling

```python
"""Config for training a decoder-only language model."""
config.seed = 0
config.total_steps = ...  # swept
config.vocab_size = 32,000

config.input = dict()
config.input.max_len = 1,024  # or 1,536 for long-sequence baseline
config.tokenizer = 'c4_en'

config.input.data = {  # equal weight
    'c4/en': 1.0,
    'huggingface:cerebras__slimpajama_627b': 1.0,
}
for dataset_name in config.input.data:
  config.input[dataset_name] = {}
  config.input[dataset_name].data = dict(
      name=dataset_name,
      split='train',
  )
  config.input[dataset_name].shuffle_buffer_size = 250,000

config.input.batch_size = 1,024

# Optimizer section
config.optax_name = 'scale_by_adam'
config.grad_clip_norm = 1.0

config.lr = 5e-4
config.wd = 5e-5
config.schedule = dict(decay_type='rsqrt',
                       warmup_steps=5,000,
                       cooldown_steps=5,000,
                       )
```

### E.2.2 Vision: Image Classification

```python
"""Config for training a ViT model."""
config.seed = 0
config.total_epochs = ...  # swept
config.num_classes = 1,000
config.init_head_bias = -6.9

config.input = dict()
config.input.batch_size = 1,024
config.input.shuffle_buffer_size = 250,000

# preprocessing
config.input.pp = 'value_range(-1, 1)|inception_crop(224)|flip_lr'
config.mixup = dict(p=0.2, fold_in=None)

# Optimizer section
config.optax_name = 'scale_by_adam'
config.grad_clip_norm = 1.0
config.optax = dict(mu_dtype='bfloat16', b2=0.95)

config.lr = 0.001
config.wd = 0.0001

config.schedule = dict(decay_type='cosine', warmup_steps=10,000)
```

### E.2.3 Language-Image Pretraining

```python
"""Config for training a SigLIP model."""
config.seed = 0
config.total_examples = ...  # swept

config.input = dict()
config.input.batch_size = 1,024 x 32
config.input.shuffle_buffer_size = 250,000

# preprocessing
config.input.tokenize = mc4  # multilingual c4
config.input.max_len = 64  # for text
config.input.prefetch = 1  # save host memory

# model
config.model.bias_init = -10.0
config.model.temperature_init = 10.0

# Optimizer section
config.optax_name = 'scale_by_adam'
config.grad_clip_norm = 1.0

config.lr = 0.001
config.wd = 0.0001

config.schedule = dict(decay_type='cosine', warmup_steps=20_000)
```

# F    C4 Evaluation Results with Stochastic RINS

Figure 9: Performance of stochastic RINS ($A^3B$) with varying inference costs for 1B parameter LMs on C4, similar to Figure 4 reported on SlimPajama. The $x$-axis represents the training compute cost. The legend indicates the inference cost of each stochastic RINS configuration relative to the baseline; e.g. $1.5x$ denotes 50% increase in inference cost. For $p_s = 0$, RINS@1x is significantly worse, with perplexity scores $> 3$. As expected, RINS converges in performance to the baseline as $p_s \to 1$.

# G  Vision

As previously discussed, the performance gains in RINS are consistent with the self-similar nature of language. By performing a recursive, scale-invariant decoding, RINS introduces an inductive bias that encourages the model to recognize and exploit recurring patterns at different scales (see Appendix D for further discussion). To test if this is likely the source of its advantage, we conduct a similar empirical evaluation in vision, a domain lacking self-similarity.

**Setup.** We train an encoder-only vision transformer ViT-B/16 Dosovitskiy et al. (2021) on ImageNet-ILSRCV2012 Deng et al. (2009). The non-recursive baseline model is trained for either 300 or 1,000 epochs using a batch size 1,024, while recursive models are trained on fewer epochs to match the same total training compute FLOPs. We apply MixUp (probability 0.2) during training Zhang et al. (2018) and use learned position embedding. The optimizer is Adam where we tune the learning rate for each architecture in the set $\mathrm{lr} \in \{10^{-3}, 7 \times 10^{-4}, 3 \times 10^{-4}, 10^{-4}\}$ with weight decay $\mathrm{wd} = \mathrm{lr}/10$, on a small validation split. We use a cosine learning rate schedule with 10K warm-up steps. Images are resized to $224 \times 224$ and $16 \times 16$ patch sizes are used. The full training configuration is in Appendix E.

**Results.** As presented in Table 7, parameter-sharing techniques, including RINS, do not confer any advantage in supervised image classification. The non-recursive baseline, when trained on longer sequence lengths (i.e., higher image resolution) to match the inference cost of recursive architectures, surpasses all other methods. This starkly differs from the results observed in language modeling, where RINS provides significant gains even when compared against models that scale inference by increasing the sequence length.

| Architecture | Val | ReaL | v2 | Avg |
|---|---|---|---|---|
| | | 300 epochs | | |
| (ABB)$_2$ | **75.2** | **81.4** | **62.7** | **73.1** |
| A@336 | **75.7** | **81.0** | **62.3** | **73.0** |
| AAB | 75.2 | 80.5 | 61.4 | 72.4 |
| ABBC | 75.1 | 80.2 | 61.3 | 72.2 |
| A@224 | 74.9 | 80.1 | 61.3 | 72.1 |
| | | 1,000 epochs | | |
| A@336 | **77.6** | **82.7** | **64.6** | **75.0** |
| ABBC | 76.3 | 81.6 | 63.2 | 73.7 |
| AABC | 76.0 | 81.4 | 62.4 | 73.2 |
| AA | 75.8 | 81.4 | 62.4 | 73.3 |
| AAA | 75.7 | 80.8 | 62.0 | 72.8 |

Table 7: Performance of the top 5 architectures on ILSRCV2012 classification. The table compares the performance of recursive architectures with a baseline trained 224- and 336-resolution images. The baseline A@336, trained on higher-resolution to match the inference cost of the recursive models, outperforms all parameter-sharing architectures. We use the notation $(\mathrm{signature})_{\mathrm{degree}}$.

