# OpenReview forum: "Recursive Inference Scaling: A Winning Path to Scalable Inference in Language and Multimodal Systems"
_NeurIPS.cc/2025/Conference — NeurIPS 2025 poster_

### Official Review · Reviewer_oAMw · 2025-06-07

**Clarity:** 2
**Significance:** 2
**Originality:** 2
**Rating:** 2
**Confidence:** 4

**Summary:**

The paper proposes Recursive INference Scaling (RINS), a recursive architectural design aimed at improving inference performance in language and multimodal models. RINS recursively applies early parts of the model (Block A) multiple times before forwarding to the later blocks (Block B), with the idea of leveraging the self-similar (fractal) structure of language. The authors benchmark this against other recursive methods like Repeat-All-Over (RAO) and show gains in both language modeling and multimodal tasks under compute-matched settings.

**Questions:**

- Would RINS still outperform simpler sampling-based ensemble methods if evaluated under equal wall-clock time constraints?

- How do you recommend partitioning a large pretrained model into A and B blocks in real applications—what are the criteria?

- Can you provide evidence that RINS remains beneficial when scaling to models >1B parameters with real deployment constraints?

**Ethical Concerns:**

["NO or VERY MINOR ethics concerns only"]

**Final Justification:**

I maintain the score.

**Limitations:**

No — The discussion on societal impact and practical deployment tradeoffs is underdeveloped. Authors should include analysis of runtime cost, carbon footprint, or fairness implications of enabling RINS, especially for multimodal systems used in global-scale platforms.

**Quality:**

2

**Strengths And Weaknesses:**

**Strengths**

- Compute-Controlled Experiments: The paper controls for training compute, which is commendable and eliminates a common confounding factor in model comparisons.

- Multimodal Evaluation: The application to SigLIP and multimodal benchmarks adds empirical breadth.

- Stochastic RINS Design: Introduces inference flexibility and reduces regret via lightweight adapters, potentially improving deployment flexibility.




**Weaknesses**

***Overhyped Novelty:***
- The idea of recursion or repeated inference is not new—RAO, latent recurrent thinking, and universal transformers have explored similar ground.

- RINS is presented as a novel and superior alternative, but it essentially tunes recursive patterns and partitions, lacking substantial conceptual innovation.

***Heavily Empirical, Light on Insight:***

- While the experiments are abundant, they mostly confirm what could be expected: more compute during inference can help, especially with training-time regularization.

- There is little theoretical grounding or mechanistic insight into why RINS specifically works better than RAO, aside from the fractal analogy.

***Limited Generalization Beyond Specific Settings:***

- The improvements are only observed in language and vision-language setups. In pure vision tasks, RINS offers no benefit—this suggests the method is brittle and task-specific.

- It is unclear if RINS scales well to larger LLMs (e.g., >7B), especially given the overhead of recursive inference.

***Deployment Realism Unclear:***

- While the paper proposes stochastic variants to mitigate inference costs, real-world deployment often requires strict latency budgets, and recursive inference is not easily compatible with that.

- The actual wall-clock inference latency and memory overhead are not reported, which is crucial for a method claiming "scalable inference".

**Lack of Simpler Baseline Comparisons:**

- Comparisons are primarily made against RAO and long-sequence baselines, but not against simpler strategies like output sampling ensembles or basic CoT prompting under same FLOPs.

- It's unclear if the gains come from architectural recursion or simply from deeper, repetitive computation paths.

---

> ### Author Rebuttal · Authors · 2025-07-27
>
> We thank the reviewer for the feedback. We hope our clarification below resolves your concerns.
>
> **Novelty**
>
> We agree that recursion is an established concept, and we have compared against prior work like RAO. In fact, our big sweep covers 59 other parameter sharing architectures. Nevertheless, we believe there is significant novelty in our work:
>
> * A strict control for training compute (FLOPs): This is a critical difference from prior work, as it allows for fair comparisons. For instance, we demonstrate that the gains from the RAO strategy in MobileLLM disappear when a non-recursive baseline is trained longer to match the same compute budget.
> * We believe we are the first to offer a **no-regret** strategy, which uses two ingredients that we propose: (1) stochastic RINS, and (2) linear adapters. This is a major contribution because with our recipe, practitioners do not lose anything by enabling RINS during training.
> * To the best of our knowledge, we are the first to study the effectiveness of recursive architectures in pure vision and vision-language settings. We demonstrate significant gains in SigLIP.
> * We carry out an extensive set of ablations. For example:
>   - we study the impact of KV cache sharing and show that even when cache sharing is used to keep memory footprint fixed, RINS still offers performance gains (Figure 6).
>   - we go beyond simple performance reporting by deriving data scaling laws for RINS, showing it improves both the scaling exponent and the irreducible asymptotic loss, proving that its benefits cannot be matched by simply training a baseline model for longer (Figure 5a).
>   - We also study the optimal number of recursion rounds against model size and training compute.
>
> In summary we do not just propose a minor variation of recursive inference. We study it rigorously on a fair compute-matched regime, resolve limitations in prior works, and compare RINS against 59 other parameter-sharing architectures.
>
>
> **Mechanistic Insight**
>
> Our hypothesis for why RINS works well in language was discussed at length in the paper using self-similarity (see for example Appendix D). Besides showing that it indeed works for language, we also present another evidence: vision (Appendix G). Since vision does not exhibit the same self-similar structure as language, we would not expect a language-inspired method to necessarily confer an advantage there, which we show to be the case indeed.
>
> We do acknowledge that rigorous theoretical proofs would be helpful but it is difficult to prove results for deep neural networks. The alternative is to treat it as a scientific phenomenon: formulate hypotheses, make predictions based on our hypotheses and test those predictions experimentally, which is what we do in the paper.
>
> **Generalization**
>
> Language is a **major** domain and introducing methods that improve language models is an important contribution. It would be unfair to require a method inspired by the beautiful self-similar structure in language to also work for another domain that doesn’t have that structure, such as vision. As we said above, the negative results in pure image classification (Appendix G) are not a contradiction but rather supporting evidence for our hypothesis regarding self-similarity.
>
> Besides, we demonstrate the success of RINS (beyond pure language) in multimodal systems where language is a key component. Our experiments with SigLIP-B/16, a vision-language model, show significant improvements across a wide range of tasks, including:
>
> * +2.3% in 0-shot ImageNet accuracy (from 77.3% to 79.6%).
> * Widespread gains in retrieval, cultural diversity, and multilinguality benchmarks (Tables 2, 3, 4, 5).
>
> **Deployment**
>
> This is an excellent point, and we are happy to provide clarification.
> We agree that latency is a limiting factor. But, memory footprint is also another major constraint (please see the discussion about why memory is a major bottleneck in Section 1 of the MobileLLM paper). For memory, a significant portion is due to the model parameters and another portion is due to the KV cache. RINS improves performance without increasing the model size. In addition, we show that it helps even when KV cache sharing is used, which keeps the memory footprint fixed.
>
> The actual wall-clock time depends on the type of hardware. To make our results more general, we compared theoretical FLOPs in the paper. Nonetheless, we report below training-time statistics on TPUv5, which we will add. to the paper for completeness:
>
> |Model Size (# params)| Recursive Signature | # examples / core / sec | Peak Memory Usage (GiB) |
> |---|---|---|---|
> |300M| Baseline | 22.7 |  2.60 |
> || RINS (A$^2$B) | 18.1  |  3.31 |
> || RINS (A$^3$B) |  13.9 |  3.56 |
> || RAO (2x) |  12.6 | 4.35  |
> |||||
> |600M| Baseline | 12.0 | 3.92 |
> || RINS (A$^2$B) | 8.5 | 5.70 |
> || RINS (A$^3$B) | 6.4 | 6.31 |
> || RAO (2x) | 6.5 | 7.48 |
>
> Note that RINS with AAB signature has a significantly better latency and memory footprint than RAO and still outperforms it significantly in terms of quality. We will add these results to the main paper.
>
> **Other Baselines**
>
> As we mention in the paper, RINS is not competing with other techniques such as CoT and repeated sampling because they can both be implemented together. In addition, CoT is specific to decoder-only architectures, while RINS can be used in encoder-only architectures, such as SigLIP.
>
> **More Inference is Better?**
>
> The claim that “more inference compute is better” so our findings aren't surprising is not accurate in our opinion. Yes, this often holds when models are trained *on the same number of tokens* but not necessarily when training compute FLOPs is controlled for. As we show in the paper, RAO and recursive inference in pure vision don’t help when we remove those confounding factors (even though we still do more inference compute at test time).
>
> **Partitioning the Model**
>
> As we discuss in Lines 151-154, we always partition the model depthwise into two equally sized blocks. It is a simple, straightforward rule that does not need further tuning.
>
> **Beyond 1B-parameter Models**
>
> As we argue from our scaling experiments, RINS helps models  with >1B parameters if they are trained on more tokens.
>
> But to further confirm this, we have trained a 2B parameter language model using either: no recursion (BL), RAO, and RINS with signature AAB. The baseline is pretrained on 200B tokens and models with recursion are trained on fewer tokens to match the total training FLOPs. As expected the baseline is always better than RAO while RINS outperforms the baseline; and the gap continues to increase in favor of RINS. This is in agreement with our scaling law studies. Please see the C4 validation perplexity scores below.
>
> Unfortunately, since we cannot attach images to the rebuttal (as per the Neurips guidelines), we provide a table of results. We hope this is sufficient to demonstrate that RINS works well even with models larger than 1B parameters. We will add this to the supplementary materials of the paper.
>
> | FLOPs (tokens x layers) | BL | RAO | RINS (A$^2$B) |
> |---|---|---|---|
> | 150K | 2.558 | 2.593 | 2.559 |
> | 250K | 2.515 | 2.534 | 2.506 |
> | 350K (after cooldown) | 2.424 | 2.437 | 2.409 |
>
>
> We hope these clarifications have addressed the reviewer's concerns and clarified the novelty and significance of our findings. We will revise the paper to make our contribution clearer and take your concerns into account. We respectfully ask the reviewer to reconsider their evaluation, and we would be happy to answer any remaining questions during the discussion period.

---

> ### Author Response · Authors · 2025-08-03
> **Follow up**
>
> Dear reviewer,
>
> Thank you again for the detailed review. We kindly would like to follow up on our rebuttal. If there are any remaining questions or concerns, please let us know so we can respond to them during the discussion period. Otherwise, we would appreciate it if you could confirm if our rebuttal has satisfactorily answered your questions.
>
> Sincerely

---

### Official Review · Reviewer_ATEM · 2025-06-30

**Clarity:** 3
**Significance:** 3
**Originality:** 3
**Rating:** 5
**Confidence:** 3

**Summary:**

Recursive Inference Scaling (RINS) work has taken inspiration from the self-similar(fractal) nature of language modality and seeks to exploit scale-invariant decoding for better generalization. The methodology decomposes a model into two sequential blocks (A & B), and then recursively applies first block A multiple times (A^rB structure). This helps in increasing the inference time depth without increasing parameter count. An exhaustive ablation and empirical study have been performed using a compute-matched evaluation framework. RINS consistently outperforms most of the baselines.
RINS also showcase better scaling exponents and asymptotic limits, demonstrating superior sample efficiency and generalization.
Further, the paper introduces stochastic RINS, where recursion depth is randomized during training, and shows that incorporating lightweight linear adapters(<1%) enables no-regret-training. RINS trained models retain improved performance even when recursive inference is skipped at test time. Additionally, KVcache sharing study has been performed, to reduce memory cost while maintaining performance gains. Overall, RINS aims to provide plug and play method for scaling inference in compact models, with benefits validated across both language and multimodal tasks.

**Questions:**

1. It would be nice to get some theoretical insights on what exactly RINS is benefitting from, or what is the recursive blocks optimizing or learning.

**Ethical Concerns:**

["NO or VERY MINOR ethics concerns only"]

**Final Justification:**

The rebuttal addresses most of the weakness and the recursion signature, degree, adapter use, KVcache sharing provide insights into the trade offs and behavior of RINS which has been proved through extensive empirical study.

**Limitations:**

yes

**Quality:**

4

**Strengths And Weaknesses:**

Strengths-
1. The paper provides a highly empirical study with evaluation on more than 55 architectural variants, multiple model sizes(300M-1B), and tasks. All experiments are compute and parameter matched, ensuring fair comparisons. It is great to see alot of common downstream tasks were covered including but not limited to OpenBookQA, PIQA, 0-shot Imagenet classification.
2. The authors have presented the method, motivation and results in a well-structured and accessible manner. The recursion signature, degree, adapter use, KVcache sharing provide insights into the trade offs and behavior of RINS. The notation, taxonomy and diagrams aid understanding.
3. The training details, datasets, FLOPs and failure cases(OOM for some models) are explicitly detailed.
4. The use case is highly relevant for deployment constrained environments.
5. The no-regret training setup, that provides training time benefits even without using recursion at test time is an important and practical innovation that lowers adoption cost.
6. The motivation from fractal hypothesis of language and the idea of recursively applying a subset of model instead of full model seems quite novel and interesting.

Weakness-
1. While the paper is extremely insightful and provides strong empirical justification, it is not clear what is the deep theoretical framework explaining why RINS work so great. It isn't very clear about what exactly the recursive A block learning or refining over iterations.
2. While inference compute(FLOPs) is matched, real world wall-time or memory benchmarks which are crucial for deployment are not presented.
3. Most of the experiments are limited to 300M-1B parameter model, would it provide gains for actual LLM use case models.

---

> ### Author Rebuttal · Authors · 2025-07-28
>
> We thank the reviewer for the detailed feedback. We are very pleased that you have found our work insightful. Please see our response to your questions below. We would be happy to answer any further questions during the discussion period.
>
> **Theoretical Analysis**
>
> Our hypothesis for why RINS works well in language is discussed at length in the paper using self-similarity (see for example Appendix D). Besides showing that RINS indeed works for language, we also present another important evidence: vision (Appendix G). Since vision does not exhibit the same self-similar structure as language, we would not expect a language-inspired method to necessarily confer an advantage there, which we show to be the case indeed.
>
> We do acknowledge that rigorous theoretical proofs would be helpful but it is difficult to prove results theoretically for deep neural networks. The alternative is to treat it as a scientific phenomenon: formulate hypotheses, make predictions based on our hypotheses and test those predictions experimentally, which is what we do in the paper.
>
> **Latency and Memory**
>
> The actual wall-clock time depends on the type of hardware. To make our results more general, we compared theoretical FLOPs in the main paper. Nonetheless, we report below training-time statistics on TPUv5, which we will add to the paper for completeness.
>
> |Model Size (# params)| Recursive Signature | # examples / core / sec | Peak Memory Usage (GiB) |
> |---|---|---|---|
> |300M| Baseline | 22.7 |  2.60 |
> || RINS (A$^2$B) | 18.1  |  3.31 |
> || RINS (A$^3$B) |  13.9 |  3.56 |
> || RAO (2x) |  12.6 | 4.35  |
> |||||
> |600M| Baseline | 12.0 | 3.92 |
> || RINS (A$^2$B) | 8.5 | 5.70 |
> || RINS (A$^3$B) | 6.4 | 6.31 |
> || RAO (2x) | 6.5 | 7.48 |
>
> Note that RINS with A$^2$B signature has a significantly better latency and memory footprint than RAO and still outperforms it significantly in terms of quality. We will add these results to the main paper.
>
> **Model Size**
>
> As we argue from our scaling experiments, RINS helps models  with >1B parameters if they are trained on more tokens.
>
> But to further confirm this, we have trained a 2B parameter language model using either: no recursion (BL), RAO, and RINS with signature AAB. The baseline is pretrained on 200B tokens and models with recursion are trained on fewer tokens to match the total training FLOPs. As expected the baseline is always better than RAO while RINS outperforms the baseline; and the gap continues to increase in favor of RINS. This is in agreement with our scaling law studies. Please see the C4 validation perplexity scores below.
>
> Unfortunately, since we cannot attach images to the rebuttal (as per the Neurips guidelines), we provide a table of results. We hope this is sufficient to demonstrate that RINS works well even with models larger than 1B parameters. We will add this to the supplementary materials of the paper.
>
> | Training FLOPs (tokens x layers) | BL | RAO | RINS (A$^2$B) |
> |---|---|---|---|
> | 150K | 2.558 | 2.593 | 2.559 |
> | 250K | 2.515 | 2.534 | 2.506 |
> | 350K (after cooldown) | 2.424 | 2.437 | 2.409 |
>
> **Summary**
>
> We hope these clarifications have addressed the reviewer's questions and strengthened our contribution. We will revise the paper to take your questions into account. We would be happy to answer any remaining questions during the discussion period.

---

> > ### Comment · Reviewer_ATEM · 2025-08-05
> >
> > This addresses most of the concern. I will maintain the score.

---

### Official Review · Reviewer_VPZK · 2025-07-03

**Clarity:** 3
**Significance:** 3
**Originality:** 3
**Rating:** 5
**Confidence:** 3

**Summary:**

This paper introduces Recursive INference Scaling (RINS), a method for scaling inference time in language and multimodal models by recursively applying a subset of model layers. The paper splits a model into two blocks A and B, where block A is applied recursively r times to its own output before passing to block B (A^r B). Further combining with stochastically dropping and linear adapter, RINS-enabled pretraining improves performance in language modeling even when recursive depth is not applied at inference time. The paper evaluates RINS against 55+ other parameter-sharing strategies under compute-matched conditions and demonstrate consistent improvements in language modeling and multimodal tasks.

**Questions:**

- Could you explain more the hypothesize about self-similar geometry and how it supports RINS theoretically.
- Why some results show very large error bars?

**Ethical Concerns:**

["NO or VERY MINOR ethics concerns only"]

**Final Justification:**

The paper proposes an elegant yet effective method, and systematically studies the scaling law of it, across model size, modalities, tasks, and data size and shows the benefits of efficiency (memory foot print and FLOPs), downstream quality, practical feasibility.

**Limitations:**

See weaknesses.

**Quality:**

3

**Strengths And Weaknesses:**

Strengthes:
- The paper proposes a simple, elegant and effective A^r B parameter-sharing method, and systematically studies the scaling law of it, across model size, modalities, tasks, and data size.
- The comparison with the control of training budget is fair and meaningful.
- Provide a stochastic RINS which improves performance even when recursion isn't used at inference, making the method more practical feasible.
- Analyze the KV cache sharing for reduced memory footprint.

Weaknesses:
- The paper targets the deployment environments with stringent memory limitations, but does not include latency analysis or profiling memory footprint.
- Evaluations are focused primarily on C4/SlimPajama and perplexity for language modeling.

---

> ### Author Rebuttal · Authors · 2025-07-28
>
> We thank the reviewer for the feedback. We are very pleased that you have found our work elegant and insightful. Please see our response to your questions below. We would be happy to answer any further questions during the discussion period.
>
> **Latency and Memory**
>
> The actual wall-clock time depends on the type of hardware. To make our results more general, we only compared theoretical FLOPs in the main paper. Nonetheless, we report below training-time statistics on TPUv5:
>
> |Model Size (# params)| Recursive Signature | # examples / core / sec | Peak Memory Usage (GiB) |
> |---|---|---|---|
> |300M| Baseline | 22.7 |  2.60 |
> || RINS (A$^2$B) | 18.1  |  3.31 |
> || RINS (A$^3$B) |  13.9 |  3.56 |
> || RAO (2x) |  12.6 | 4.35  |
> |||||
> |600M| Baseline | 12.0 | 3.92 |
> || RINS (A$^2$B) | 8.5 | 5.70 |
> || RINS (A$^3$B) | 6.4 | 6.31 |
> || RAO (2x) | 6.5 | 7.48 |
>
> Note that RINS with A$^2$B signature has a significantly better latency and memory footprint than RAO and still outperforms it significantly in performance. We will add these results to the main paper for completeness.
>
> **Evaluation**
>
> While we do focus on perplexity (PPL) during the big sweep (where we compare 59 parameter-sharing architectures), we also evaluate RINS against popular baselines in downstream tasks. This includes:
>
> * For language models, zero-shot evaluation on common sense reasoning tasks (Table 1).
> * For SigLIP, zero-shot classification, retrieval, and cultural diversity metrics (Tables 2, 3, 5) and multilinguality (Appendix B, Table 4).
> * Analysis of how RINS improves both the scaling exponent and the asymptotic performance limits (Section 5, Figure 5a).
>
> In addition, we will also include the latency and memory footprint results above in the main paper for completeness.
>
> **Recursive Inference and Self-Similarity**
>
> We provide more details about it in Appendix D. Informally, self-similarity means that similar patterns exist at multiple scales in language (e.g. across paragraphs vs. across larger scopes). Since deeper layers encode high-level contextual information (which correspond to higher-level representations), we expect that a portion of what the early layers learn can be directly used in higher layers as well. But this is precisely what recursive inference enables.
>
> **Variance in Results**
>
> A few downstream evaluations have a large variance. In BoolQ, for example, if you compare Table 10 in MobileLLM paper, the 350M baseline gets 53.8% while the much smaller 125M baseline gets 57.5%, despite the fact that larger models tend to do better overall. This is most probably due to variance. Interestingly, recursive inference mitigates this noise as seen in Table 1 in our paper.
>
> **Summary**
>
> We hope these clarifications have addressed the reviewer's questions. We will revise the paper to make our contribution clearer and take your questions into account. If our answers are satisfactory, we respectfully ask the reviewer to update their score, and we would be happy to answer any remaining questions during the discussion period.

---

> ### Comment · Reviewer_VPZK · 2025-08-06
>
> Thank you for the detailed response. You have addressed all my concerns. I have raised my score.
>
> The paper would be benefit to show the real latency improvement, and more (theoretical or intuitive) insights on RINS in the revision.

---

### Official Review · Reviewer_vR9m · 2025-07-05

**Clarity:** 2
**Significance:** 2
**Originality:** 2
**Rating:** 4
**Confidence:** 2

**Summary:**

This paper proposes that the (A^r)B recurrent structure is the best structure for parameter sharing. However, I find several parts of the article confusing and difficult to understand. I hope the author can clarify these points during the rebuttal phase.

**Questions:**

My questions are listed above.

**Ethical Concerns:**

["NO or VERY MINOR ethics concerns only"]

**Final Justification:**

The rebuttal by authors addresses most of my questions. The paper effectively demonstrates the empirical advantages of RINS, showcasing its strong performance across various experimental settings. However, it would benefit from deeper insights into why RINS functions as the most effective structure.

**Limitations:**

YES

**Quality:**

3

**Strengths And Weaknesses:**

I list some of the points that I find confusing as follows:
- Why it is a winning path to scalable inference in Language and **Multimodal Systems**, regarding that this paper (Appendix G) concludes that "parameter-sharing techniques, including RINS, do not confer any advantage in supervised image classification. The non-recursive baseline surpasses all other methods. This starkly differs from the result observed in language modeling." Though Section 6 provides some other experiments about multimodality. It is still unclear what the relationship between these two parts is.
- How is this paper related to **inference scaling**? The experiments focus on the PPL and training FLOPs. I don't find contents related to scalable inference.
- The experiments are conducted on 300M/600M models. We can hardly say it is a **winning path** based on observations from such small models.

---

> ### Author Rebuttal · Authors · 2025-07-26
>
> We thank the reviewer for the feedback. We believe there is a misunderstanding regarding the scope and key results of our paper, and we hope our clarification below will resolve it.
>
> **The Scope of "Multimodal Systems" and the Role of Vision Experiments**
>
> We suspect that the reviewer is mixing the term “multimodal systems” with the term “all modalities”. Our claim is that RINS is effective for language and multimodal systems that process language (such as contrastive vision-language models). We never claimed that it works for all modalities.
>
> Language is a **major** domain and introducing methods that improve language models is an important contribution. It would be unfair to require a method inspired by the beautiful self-similar structure in language to also work for another domain that doesn’t have that structure, such as vision.
>
> The negative results in pure image classification (Appendix G) are not a contradiction but rather supporting evidence for our hypothesis. Since vision as a modality does not exhibit the same self-similar structure as language, we would not expect a language-inspired method to necessarily confer an advantage in pure vision.
>
> We demonstrate the success of RINS in multimodal systems where language is a key component. Our experiments with SigLIP-B/16, a vision-language model, show significant improvements across a wide range of tasks, including:
> * +2.3% in 0-shot ImageNet accuracy (from 77.3% to 79.6%).
> * Widespread gains in retrieval, cultural diversity, and multilinguality benchmarks (Tables 2, 3, 4, 5).
>
> These results strongly support our claim that RINS is a "winning path" for systems that process natural language, whether in isolation or as part of a multimodal architecture. We say it is a "winning path" because we have compared it against 59 other parameter-sharing architectures.
>
> **The Connection to "Inference Scaling"**
>
> RINS is, by definition, an inference-time scaling method. RINS scales the amount of computation at inference time by recursively applying a subset of the model's layers (e.g., using signature $A^rB$). This is achieved while keeping the model parameter count and the total  training compute (FLOPs) fixed relative to baselines.
>
> **The Evaluation and PPL**
>
> Our evaluation is comprehensive and not limited to perplexity (PPL). This includes:
>
> * For language models, zero-shot evaluation on common sense reasoning tasks (Table 1).
> * For SigLIP, zero-shot classification, retrieval, multilinguality, and cultural diversity metrics.
> * Analysis of how RINS improves both the scaling exponent and the asymptotic performance limits (Section 5, Figure 5a).
>
> **The Scale of Models Used in Experiments**
>
> While our initial, broad sweep of 59 different architectures was conducted on 300M and 600M parameter models for computational feasibility, we validated our key findings on larger 1B parameter models. See for example: Figure 3b, Figure 4, Figure 5b. Also, as we argue from our scaling experiments in Figure 5b, RINS helps models  with >1B parameters if they are trained on more tokens.
>
> But to verify this more, we have now trained a 2B parameter language model using either: no recursion (BL), RAO, and RINS with signature AAB. The baseline is pretrained on 200B tokens and models with recursion are trained on fewer tokens to match the total training FLOPs. As expected the baseline is always better than RAO while RINS outperforms the baseline; and the gap continues to increase in favor of RINS. This is in agreement with our scaling law studies. Please see the C4 validation perplexity scores below.
>
> Unfortunately, since we cannot attach images to the rebuttal (as per the new Neurips guidelines), we provide a table of results. We hope this is sufficient to demonstrate that RINS works well even with models >1B parameters. We will add this to the supplementary materials of the paper.
>
> | Training FLOPs (tokens x layers) | BL | RAO | RINS (AAB) |
> |---|---|---|---|
> | 150K | 2.558 | 2.593 | 2.559 |
> | 250K | 2.515 | 2.534 | 2.506 |
> | 350K (after cooldown) | 2.424 | 2.437 | 2.409 |
>
>
>
> **Summary of Key Contributions**
>
> We feel it is unfortunate that most of our contributions that distinguish our work from others seem to have been overlooked in this review, including :
>
> 1. *No-Regret" Strategy*: We introduce stochastic RINS and show that stochastic RINS with linear adapters offer a **no-regret** strategy. This is a major contribution because using our recipe, one can choose not to use recursive inference at test time with no loss in performance while still having the option to use recursive inference to improve performance significantly *for the same set of weights and same training FLOPs*.
>
> 2. *Multimodal and Vision Contexts*: To our knowledge, this is the **first** work to systematically study the effectiveness of recursive architectures in both pure vision and vision-language settings. We demonstrate significant gains in SigLIP.
>
> 3. *Memory-Efficient Inference*: We study the impact of KV cache sharing and show that even when cache sharing is used to keep memory footprint fixed, RINS still offers performance gains (Figure 6).
>
> 4. *Scaling Law Analysis*: We go beyond simple performance reporting by deriving data scaling laws for RINS, showing it improves both the scaling exponent and the irreducible asymptotic loss, proving that its benefits cannot be matched by simply training a baseline model for longer (Figure 5a). We also study the optimal number of recursion rounds against model size and training compute.
>
> We hope these clarifications have addressed the reviewer's concerns and clarified the novelty and significance of our findings. We will revise the paper to make our contribution clearer and take your concerns into account. We respectfully ask the reviewer to reconsider their evaluation, and we would be happy to answer any remaining questions during the discussion period.

---

> > ### Comment · Reviewer_vR9m · 2025-08-02
> > **Response to Rebuttal**
> >
> > Thanks for the clarifications. They address most of my questions. I raise the score.

---

### Note · Authors · 2025-08-13

Dear AC and reviewers,

Thank you for your time and efforts. We would like to summarize our rebuttal below for your convenience:

* We have added new experiments on 2B-parameter language models, which show that our conclusions still hold. RINS offers a significant advantage compared to other parameter-sharing architectures. We'll add these new results to the supplementary materials of the paper.

* We have added wall-clock time and memory footprint statistics. These show that RINS has an advantage over RAO (used in MobileLLM): much better performance with less latency and memory footprint.

* Our work is different from prior works because:

  - We control for training compute (FLOPs). This is a critical difference from prior work. For instance, we demonstrate that the gains from the RAO strategy in MobileLLM disappear when a non-recursive baseline is trained longer to match the same compute budget.
  - We are the first to offer a no-regret strategy through a combination of stochastic RINS and linear adapters.  This is a major contribution because with our recipe, practitioners do not lose anything by enabling RINS during training.
  - We are the first to study the effectiveness of recursive architectures in pure vision and vision-language settings. We demonstrate significant gains in SigLIP, such as an improvement in 0-shot ImageNet classification from 77.3% to 79.6%.
  - We carry out an extensive set of ablations, including using KV cache sharing, and deriving data scaling laws, and studying the optimal number of recursion rounds against model size and training compute.

We hope this clarifies our contribution. We will update the paper to incorporate all comments raised by the reviewers.

Sincerely

---

### Decision · Program_Chairs · 2025-09-17

**Decision:**

Accept (poster)

**Comment:**

This paper proposes a framework that recursively applies a subset of a model to scale inference time in LLMs and multimodal LLMs, supported by extensive empirical results. Initial reviews were polarized, and even after rebuttal and discussion, the reviewers did not reach full consensus. Most reviewers appreciated the elegance of the key idea and the strength of the results. One reviewer continued to recommend rejection without providing convincing justification or engaging with the rebuttal despite repeated reminders. After carefully considering the paper, reviews, and discussion, the AC concurs with the majority and recommends acceptance. The authors are encouraged to incorporate the rebuttal clarifications into the camera-ready version.